# HotSPOT: A Computational Tool to Design Targeted Sequencing Panels to Assess Early Photocarcinogenesis

**DOI:** 10.3390/cancers15051612

**Published:** 2023-03-05

**Authors:** Sydney R. Grant, Spencer R. Rosario, Andrew D. Patentreger, Nico Shary, Megan E. Fitzgerald, Prashant K. Singh, Barbara A. Foster, Wendy J. Huss, Lei Wei, Gyorgy Paragh

**Affiliations:** 1Department of Cell Stress Biology, Roswell Park Comprehensive Cancer Center, Buffalo, NY 14203, USA; 2Department of Dermatology, Roswell Park Comprehensive Cancer Center, Buffalo, NY 14203, USA; 3Department of Biostatistics and Bioinformatics, Roswell Park Comprehensive Cancer Center, Buffalo, NY 14203, USA; 4Department of Genetics and Genomics, Roswell Park Comprehensive Cancer Center, Buffalo, NY 14203, USA; 5Department of Pharmacology and Therapeutics, Roswell Park Comprehensive Cancer Center, Buffalo, NY 14203, USA

**Keywords:** genomics, next generation sequencing, hotspots, mutations, carcinogenesis, cutaneous squamous cell carcinoma, clonal mutations

## Abstract

**Simple Summary:**

Mutations are present in healthy skin long before clinical signs of skin cancer arise. Many studies have shown that mutations in healthy tissue and cancer cluster at specific areas in the genome, often referred to as mutation hotspots. Next-generation sequencing has become the gold standard for studying cancer genomics. However, it is not economically feasible to sequence large genomic regions at the depth necessary to study mutations in healthy tissues. We have created an algorithm that formats mutation data into a targetable panel of genomic segments that can be used to design sequencing experiments. The efficacy of our algorithm was tested using three publicly available datasets. Compared to the original genomic regions used for these studies, the regions identified by our algorithm improved mutation capture efficacy ranging from 9.6 to 12.1-fold. Our web application hotSPOT provides a publicly available resource for researchers to design next-generation sequencing experiments to effectively study mutations in healthy tissues and cancer.

**Abstract:**

Mutations found in skin are acquired in specific patterns, clustering around mutation-prone genomic locations. The most mutation-prone genomic areas, mutation hotspots, first induce the growth of small cell clones in healthy skin. Mutations accumulate over time, and clones with driver mutations may give rise to skin cancer. Early mutation accumulation is a crucial first step in photocarcinogenesis. Therefore, a sufficient understanding of the process may help predict disease onset and identify avenues for skin cancer prevention. Early epidermal mutation profiles are typically established using high-depth targeted next-generation sequencing. However, there is currently a lack of tools for designing custom panels to capture mutation-enriched genomic regions efficiently. To address this issue, we created a computational algorithm that implements a pseudo-exhaustive approach to identify the best genomic areas to target. We benchmarked the current algorithm in three independent mutation datasets of human epidermal samples. Compared to the sequencing panel designs originally used in these publications, the mutation capture efficacy (number of mutations/base pairs sequenced) of our designed panel improved 9.6–12.1-fold. Mutation burden in the chronically sun-exposed and intermittently sun-exposed normal epidermis was measured within genomic regions identified by hotSPOT based on cutaneous squamous cell carcinoma (cSCC) mutation patterns. We found a significant increase in mutation capture efficacy and mutation burden in cSCC hotspots in chronically sun-exposed vs. intermittently sun-exposed epidermis (*p* < 0.0001). Our results show that our hotSPOT web application provides a publicly available resource for researchers to design custom panels, enabling efficient detection of somatic mutations in clinically normal tissues and other similar targeted sequencing studies. Moreover, hotSPOT also enables the comparison of mutation burden between normal tissues and cancer.

## 1. Introduction

Genetic mutations are acquired throughout the lifespan in human tissues. Most of these mutations will slowly accumulate and result in no observable architectural or functional changes. However, some early mutations in oncogenes or tumor suppressor genes may preferentially allow cell clones to acquire other mutations and initiate carcinogenesis [1]. A better understanding of early carcinogenesis can aid cancer risk assessment and help develop better prevention and treatment strategies. 

Sequencing has become very affordable in recent years, with whole exome or whole genome sequencing being more widely available when studying the cancer genome. However, ultra-deep sequencing of a very high sample number is required to establish the pattern of early clonal mutations in normal skin, which is only feasible by targeting finite genomic regions. Previous studies on early clonal mutations in normal skin have sequenced panels of exomes of genes frequently mutated in cancer [2,3,4]. Given the non-random distribution of mutations, this approach is inherently inefficient. 

Acquired genetic mutations are often a result of carcinogen exposure. One of the best-known human carcinogens is ultraviolet B (UVB) light [5]. Driver mutations in many frequently mutated skin cancer genes (*TP53*, *CDKN2A, FAT1, NOTCH1,* and *NOTCH2*, etc.) bare a UV-signature pattern in cutaneous squamous cell carcinomas (cSCC) [6]. Mutations are not only present in late UV carcinogenesis. Decades before clinical evidence of disease, the human skin is already littered with somatic mutations induced by UV, other environmental exposure, and aging [1]. Rather than occurring randomly, different genomic areas show variable susceptibility to carcinogen-induced DNA lesions and have disparate repair activity [7]. Moreover, the cellular effect of mutations also makes some mutations more likely to propagate in tissues while others are more likely to be lost. Based on the non-random pattern of UV-induced mutations in keratinocytes and cSCC and our own limited initial normal skin sequencing data [2,3,4,6,8,9], we hypothesized that mutations in normal skin also cluster around hotspots and these hotspots can help design more efficient targeted sequencing panels. However, we found no software tools to identify and compare mutational hotspots for targeted sequencing panel design. 

Previously, other groups have begun to test the efficacy of targeted sequencing panels designed based on whole-exome regions likely to be mutated [10]. Additionally, several groups have developed computational resources to find hotspot regions in cancer [11,12,13,14,15]. However, published resources still use large genomic windows and cannot be customized to individual sequencing experiment needs. Currently, there is no publicly available tool to design optimal library preparation targets of the most mutated genomic regions for high-depth targeted sequencing and to compare these targeted hotspot areas. Inclusion of infrequently mutated areas in sequencing panels can markedly increase cost without sufficiently improving the power of comparing mutational load between different tissues. Thus, more efficient targeting of frequently mutated areas was sorely needed for studies focusing on early mutational patterns. 

Our work describes hotSPOT, our easy-to-use software for identifying optimal targeted sequencing panels for high-depth sequencing projects and comparing mutational patterns. Moreover, using hotSPOT, we show the remarkable clustering of mutations in sun-exposed normal skin, and we offer novel evidence for the overlap between normal skin and cSCC mutations. Finally, we demonstrate the ability of this software to compare mutational burden between samples by showing the mutational differences between a clinically normal epidermis with chronic and intermittent sun exposure history. 

## 2. Materials and Methods

### 2.1. Datasets

To test the utility of our computational tool, we used three independent datasets depicting point mutations in clinically normal-appearing skin. The datasets used for this study varied in sample size, sequencing depth, sequencing panel size, and location from which samples were taken. The sequencing parameters of these datasets were summarized (Table 1) and considered for all comparative analyses. Dataset A from Fowler et al. sequenced 1261 epidermal samples from 35 individuals undergoing either melanoma excision, cosmetic surgery, or deceased organ donors. Epidermal samples were collected from varying sites, including the head, trunk, forearm, abdomen, and leg. Further, 2 mm^2^ regions of normal epidermis were used for targeted sequencing of 74 known cancer genes [3]. Dataset B from Martincorena et al. included a total of 234 biopsies from the sun-exposed eyelid epidermis of four individuals undergoing cosmetic surgery. Biopsies ranged from approximately 0.8 to 4.7mm^2^ in area and were used for targeted sequencing of 74 known cancer genes [4]. Dataset C from Hernando et al. included 123 epidermal biopsies from 123 individuals taken from the margins of skin excision biopsies for the removal of benign lesions. Epidermal samples were collected from varying sites, including the back, chest, legs, arms, neck, face, and hands. The size of samples varied based on the size of the lesion, with surgical margins ranging from 1 to 3 mm [2]. Additionally, two datasets were utilized, including cSCC tissue samples from 39 [6] and 36 [9] patients.

### 2.2. Sequencing Panel Identifier Input Data and Amplicon Generator

hotSPOT requires input that includes: chromosome and base pair location of mutations in the tissue of interest, the library capture assay’s optimal amplicon length, and the size of desired output sequencing panel. To optimize the identification of the most frequently mutated genomic targets with a low computational footprint algorithm, we tested multiple iterations of the algorithm and employed the following as the final design. 

This algorithm searches the mutational dataset (input) for mutational hotspot regions on each chromosome:1.Starting at the mutation with the lowest chromosomal position (primary mutation), using a modified rank and recovery system, the algorithm searches for the closest neighboring mutation.2.If the neighboring mutation is less than one amplicon in distance away from the primary mutation, the neighboring mutation is included within the “hotspot” region.This rank and recovery system is repeated, integrating mutations into the “hotspot region” until the neighboring mutation is greater than or equal to the length of one amplicon in distance, from the primary mutation (Figure 1A, Appendix A). Once neighboring mutations equal or exceed one amplicon in distance from the primary mutation, incorporation into the “hotspot region” halts incorporation.3.For hotspots within the one amplicon range, from the lowest to highest mutation location, this area is covered by a single amplicon and added to an amplicon pool with a unique ID.The center of these single amplicons is then defined by the weighted distribution of mutations. 4.For all hotspots larger than one amplicon, the algorithm examines 5 potential amplicons at each covered mutation in the hotspot:one amplicon directly upstream of the primary mutationone amplicon directly downstream of the primary mutationone amplicon including the mutation at the end of the read and base pairs (amplicon length 1) upstreamone amplicon including the mutation at the beginning of the read and base pairs (amplicon length 1) downstreamone amplicon with the mutation directly in the center (Figure 1A, Appendix A).5.All amplicons generated for each hotspot region of interest are assigned a unique ID and added to the amplicon pool (Figure 1A, Appendix A).

### 2.3. Forward Selection Sequencing Panel Identifier (Optimal computation time)

6.Amplicons covering hotspots less than or equal to one amplicon in length are added to the final sequencing panel dataset.7.For amplicons covering larger hotspot regions, the algorithm uses a forward selection method to determine the optimal combination of amplicons to use in the sequencing panel:the algorithm first identifies the amplicon containing the highest number of mutationsthe algorithm then identifies the next amplicon, which contains the highest number of new mutationsthis process continues until all mutations are covered by at least one amplicon (Figure 1B, Appendix A)8.Each of these amplicons are then added to the final sequencing panel, with their own unique IDs.9.All amplicons in the final sequencing panel are ranked from highest to lowest based on the number of mutations they cover.10.The algorithm then calculates the cumulative base-pair length and the cumulative mutations covered by each amplicon.11.Dependent on the desired length of the targeted panel, a cutoff may be applied to remove all amplicons which fall below a set cumulative length (Figure 1B, Appendix A).

### 2.4. Comprehensive Selection Sequencing Panel Identifier (Optimal Mutation Capture)

12.To conserve computational power, the forward selection sequencing panel identifier is run to determine the lowest number of mutations per amplicon (mutation frequency) that need to be included in the predetermined length sequencing panel (Figure 1B, Appendix A).any amplicon generated by the algorithm which is less than this threshold value will be removed (Appendix A).13.For the feasible exhaustive selection of amplicon combinations covering hotspot areas larger than the predefined number of amplicons in length, the algorithm breaks these large regions into multiple smaller regions.
The amplicons covering these regions are pulled from the amplicon pool based on their unique IDs.14.The algorithm finds both the minimum number of amplicons overlap and all positions with this value and identifies the region with the longest continuous spot of minimum value.
The region is split at the center of this longest continuous minimum post values and continues the splitting process until all smaller regions are less than the “n” number amplicon length set by the user (Appendix A).As this set number of amplicons decreases, the computation time required also often decreases.15.All amplicons contained in these bins are added back to the amplicon pool based on a new unique ID.16.Amplicons covering hotspots less than or equal to one amplicon length are added to the final sequencing panel dataset.17.To determine the optimal combination of amplicons for each region, the number of amplicons necessary for full coverage of the bin is calculated.18.A list is generated of every possible combination of n, number of amplicons, needed. For each combination of amplicons:amplicons that would not meet the threshold of unique mutations are filtered out, and the number of all mutations captured by these amplicons is calculated.the combination of amplicons that yields the highest number of mutations is added to the final sequencing panel (Figure 1C, Appendix A).19.All amplicons in the final sequencing panel are ranked from highest to lowest based on the number of mutations they cover.20.All amplicons capturing the number of mutations equal to the cutoff are further ranked to favor amplicons that have mutations closer in location to the center of the amplicon.21.Cumulative base-pair length and cumulative mutations covered by each amplicon are calculated.Depending on the desired length of the targeted panel, a cutoff may be applied to remove all amplicons which fall below a set cumulative length (Figure 1C, Appendix A).

### 2.5. Calculation of Mutation Capture Efficacy

The mutation capture efficacy of 10,000 bp length hotSPOT designed panel was compared to that of the capture efficacy of the originally targeted regions of the published sequencing panels for each normal epidermis and cSCC dataset. One dataset for normal epidermis [3] and cSCC [6] was split 80%/20% into training and test datasets. Additional datasets were utilized as validation [2,4,9]. The normal epidermis and cSCC training datasets were modified to contain only mutations contained within the genomic regions sequenced by all compared datasets. The two training datasets were each run through the hotSPOT application to generate a 10,000 bp length sequencing panel consisting of 80,125 bp length amplicons. The total number of mutations detected per sample by the original analysis was compared to the number of these mutations which fell within the regions of the hotSPOT-designed panel. The measured mutation frequency was then normalized by the number of base pairs contained within each sequencing panel. Change in mutation capture efficacy was measured using paired Wilcox signed-rank test.

### 2.6. Computational Development of hotSPOT Algorithm and RShiny Web Application

All computational development and analyses were conducted in R version 4.1.1 [16]. Packages utilized for sequencing panel identifiers include dplyr [17], hash [18], rlist [19], and R.utils [20]. Figures were generated using ggplot2 [21], ggExtra [22], and ggpubr [23] R packages. The web application was developed using shiny [24], shinydashboard [25], shinycssloaders [26], dashboardthemes [27], DT [28], and plotly [29] R packages. 

### 2.7. Statistical Analysis

All statistical analyses were conducted in R version 4.1.1 ‘stats’ package [16]. Statistical significance was calculated using a paired Wilcox signed-rank test. Mutation distributions were compared using two-sample Kolmogorov–Smirnov tests. Each normal epidermis dataset was randomly split 50%/50% into two separate datasets for comparison of mutation distribution. Individually for each chromosome, the mutation positions of each data subset were compared using a two-sample Kolmogorov–Smirnov test. This test was repeated 100 times for 100 different random splits of each dataset. The average *p*-value for each chromosome was calculated. 

### 2.8. Calculation of Optimal Sample Size for hotSPOT Panel Design

Each normal epidermis dataset [2,3,4] and one cSCC dataset [6] were randomly split 80%/20% into training and test datasets. For all datasets, we randomly chose a subset of samples (that ranged from n = 1 sample to the entirety of the study population) and used the hotSPOT algorithm to design a targeted sequencing panel. We then calculated the change in capture efficacy of the designed panel for each sample within the test dataset compared to the original analysis. The statistical significance of mutation capture increase was calculated using a Wilcox signed-rank test. For each sample size, this test was run 100 separate times, and average *p*-values were calculated.

## 3. Results and Discussion

Mutational hotspots are well-known in cancer [30,31]. UV light also causes mutations in preferential genomic areas, but mutational hotspots in normal skin have not previously been systemically evaluated [32]. To assess whether mutational hotspots in normal skin are present and reproducible between different sequencing datasets, we compared normal skin mutation datasets A [3] and B [4]. Dataset C [2] was excluded from this analysis due to the smaller genomic region used for the original sequencing panel. To allow for a comparison of the datasets, we limited the analysis to the genomic areas sequenced in both datasets. To illustrate an example of the reproducibility of densely mutated genomic regions among different datasets, we selected two genes frequently mutated in skin cancer (*TP53, NOTCH1*) and one gene less frequently mutated in skin cancer (*RB1*). As anticipated, mutation density peaks are seen in all three genes and are less prominent in less frequently mutated genes. Representative graphs (Figure 2A) indicate the density of mutations that show overlap between the two datasets. Dataset A has over five times the number of samples and almost two times the sequencing depth of Dataset B, leading to better-defined density peaks. Based on the known presence of mutational hotspots in skin cancer, actinic keratoses, and UV-exposed keratinocytes [32], the significant overlap of mutational peaks and nadirs in normal skin was not surprising. Nevertheless, the magnitude of the mutation frequency at the hotspots vs. the background highlighted the importance of considering the variable mutation density of genomic areas during sequencing target design in projects where the efficient measuring of mutational burden differences is crucial. To further confirm the reproducibility of mutation distribution amongst epidermal samples, a two-sample Kolmogorov–Smirnov test was used to compare mutations on each chromosome. The average *p*-value calculated over 100 comparison combinations for each dataset shows no evidence suggesting a statistically significant difference in mutation distribution amongst different samples of Datasets A–C.

In previous attempts to increase the yield of high-depth sequencing, some groups have sequenced whole cancer-related genes. Other groups, including our own, have used arbitrary amplicon designation, during which the genome is consecutively broken into amplicons, and segments that captured the most mutations in prior similar datasets are selected based on predefined target capture panel size [2,3,4,8]. Although random amplicon designation markedly improves panel efficiency compared to targeting whole genes, it is still suboptimal: due to the narrow clustering of mutations in many human genomic mutation hotspots, arbitrarily assigned segment boundaries may break hotspots into neighboring amplicons. This results in under-calling the mutation density of a hotspot and at the same time over-calling the size of the hotspots, thus yielding inherent inefficiency. To find a more efficient method for developing these panels, we developed the current hotSPOT algorithm. Our first algorithm used a random amplicon binning method, breaking up the genome into 100 bp amplicons using a rank and recovery system based on how many mutations they captured. To target the algorithm toward the hotspot areas, we developed a new algorithm that grouped clusters of mutations together. This method captured more mutations. However, hotspot regions were of varying lengths and required additional processing to fit these regions into the amplicon sizes needed for sequencing panels. Because of these issues, we developed the hotSPOT algorithm as an efficient and computationally feasible way of capturing the most mutated genomic regions based on user-defined characteristics. We used Datasets A–C with a range of sequencing panel lengths to test the efficacy increase of our final hotSPOT algorithm compared to our initial conventional random amplicon designation algorithm. Across all panel lengths and datasets, the hotSPOT algorithm achieved better performance compared to our previous “conventional” method of amplicon design with an average 10.28% increase in mutation frequency (Figure 2B). These findings show the ability of the hotSPOT algorithm to identify target areas covering frequently mutated genomic regions and demonstrate significant outperformance over the previous conventionally used methods for sequencing target identification.

To test the efficacy of hotSPOT, we applied this algorithm to all three datasets to capture the ideal sequencing panel by incrementally increasing the length of the designed panel to 100,000 bp length (Figure 2C). We then measured and compared the number of mutations captured in the panels. For an unbiased comparison of mutation capture efficacy, we normalized the number of mutations captured in each section by the total mutations captured by that dataset. Additionally, as some sections cover a smaller area, the count for each section was calculated to represent the number of mutations per 10,000 bps in the datasets. The capture efficacy decreased sharply with the expansion of the target area in all datasets. To better understand the variability of hotspot regions between differing datasets, we compared the 10,000 bp panel design between all three datasets. We calculated that 20.42% of the 10,000 bp panels overlapped between all datasets (Appendix A). These data suggest that the clustering of mutational hotspots driving clonal cell growth in normal skin is similar to that of some highly mutated cancers [6,30]. As the cost of ultra-high depth sequencing is mainly defined by the size of the targeted sequencing panel [8], these data also provide evidence for considering focused ultra-high depth sequencing to keep sequencing costs at a feasible level while increasing sample size.

We additionally tested the computational optimization of both our forward and comprehensive hotSPOT algorithms compared to an exhaustive binning method. To test this exhaustive algorithm, we used the same technique as hotSPOT to identify mutation hotspots within the dataset. Then, using a sliding window of 1 bp, we generated a pool of amplicons covering every possible hotspot section. Every possible combination of amplicons was tested for total mutation capture without breaking apart any large hotspots as we do in our hotSPOT algorithm. We generated a small, targeted sequencing panel using these algorithms on four different genes in Dataset B [4] to compare computation time and mutation capture (Table 2). Our hotSPOT comprehensive (47.9- to 3.2-fold) and hotSPOT forward (460.1- to 12.0-fold) algorithms performed remarkably faster than the exhaustive algorithm. With the hotSPOT algorithm, greater speed reduction was seen in the case of more densely mutated genes. There were no differences in the identified optimal panel efficiency between the algorithms in the tested genes. We anticipate minimal differences in mutation capture for larger datasets but even more pronounced decreases in the computation time using our hotSPOT algorithms.

We then assessed the improvement of mutation capture efficacy compared to previously used targeted sequencing panels. We found an increase in average capture efficacy for all datasets, ranging from 12.1- to 9.6-fold (Figure 3A). We found the increase in capture efficacy to be statistically significant in our training (*p* < 0.001), test (*p* < 0.05) and validation #2 (*p* < 0.001) datasets, which contained the largest numbers of subjects (Figure 2E). The ability of hotSPOT to improve sequencing design efficiency was clearly demonstrated, and the reproducibility among several datasets also provided further proof for the universality of the mutational hotspot areas in normal skin.

Based on the success of mutation capture in clinically normal skin using our hotSPOT algorithm, we predicted this model could also be useful in cancer. We assessed the ability of hotSPOT to measure the mutation capture efficacy in cSCC tumor samples using two publicly available datasets [6,9]. We used hotSPOT to design a 10,000 bp targeted sequencing panel based on the training dataset. We then measured the mutation capture efficacy of our targeted sequencing panel for both the training, test, and validation cohorts. Using this targeted panel, we achieved a statistically significant increase in capture efficacy for all datasets compared to the original whole-exome sequencing (Figure 3B).

To demonstrate the ability of hotSPOT targeted panels to measure progression of carcinogenesis, the frequency of mutations in Dataset C from chronically sun-exposed (face, neck, or hands) and intermittently sun-exposed normal skin (back, chest, legs, or arms) [2] was measured within a cSCC 10,000 bp targeted sequencing panel. There was significantly higher mutation capture efficacy in chronically sun-exposed skin site compared to intermittently sun-exposed (*p* = 0.0000186) (Figure 3C). The increase in mutation burden based on history of sun-exposure shows the potential for hotSPOT identified genomic regions to serve as markers of photocarcinogenic progression. Despite multiple publications of DNA-sequencing data in normal epidermis, there are many genomic regions frequently mutated in cSCC which have not yet been studied in normal epidermis. 

To assist in the design of experiments using hotSPOT, the optimal sample size needed in the input mutation dataset was calculated for other researchers who may use hotSPOT to study their tissue of interest. We calculated this number for four different datasets of varying sequencing depths [2,3,4,6]. For average sequencing depths of 923, 690, 374, and 115x, statistical significance (*p* < 0.01) was achieved using greater than or equal to four [2], three [3], seven [4], and 18 [6] samples, respectively (Figure 4).

The visual representation of web applications for forward (Appendix A) and comprehensive (Appendix A) binning algorithms indicate data input parameters and output figures. Forward binning selection requires the input of a formatted mutation dataset (Appendix A), sequencing panel length, and amplicon length. Gene names may be entered or excluded based on availability and user preference. Comprehensive binning requires the same input parameters in addition to the size of hotspots, which indicates at which size, hotspots will be split. Both application types output a capture summary, indicating average mutations captured per amplicon, the genes captured within the amplicon, and a graphical representation of mutations captured per amplicon in the panel. The table of amplicon locations for the sequencing panel is displayed below, and the panel may be downloaded as a .csv file.

Previously, there was no tool to simplify designing sequencing panels fitted to genomic hotspot areas. Certain genomic regions are more sensitive to carcinogen exposure, leading to patterns of mutation clustering [31,32]. The predictability of these mutation sites makes this an optimal target for capturing tissue mutation burden. Deep sequencing provides a unique window into early carcinogenesis. Increasing the depth of sequencing causes a significant increase in the cost of sequencing, making whole genome and whole exome deep sequencing currently unfeasible. Because mutations often cluster around genomic hotspots, targeted sequencing panels for ultra-high depth sequencing of healthy tissues have been created by several [2,3,4,8]. These previously utilized sequencing panels have analyzed large regions of genes. While many areas of these genes are frequently populated with mutations, there are also large areas within these genes where mutations are infrequent. Identification of these genomic regions likely to be highly mutated will allow for creation of more effective targeted sequencing panels.

HotSPOT identifies and ranks genomic regions based on mutation density. However, further work must be done to identify regions both mutated and under positive selection for clonal expansion. Researchers using hotSPOT may choose not to include synonymous mutations in their input dataset, as this may lead the algorithm to identify regions which are mutated, but not relevant to carcinogenesis. Additionally, further studies on hotspots are needed to identify which regions are most useful for the assessment of disease risk, progression, and outcome.

## 4. Conclusions

We have shown mutational hotspots are present in normal skin and preserved between different normal skin sequencing datasets. Moreover, we developed a computational tool for designing efficient sequencing target panels. With the increased routine use of next-generation sequencing technologies for both clinical and research applications, there is a need for identifying which genomic regions are most relevant for studying diseases. Our web application, hotSPOT, provides a publicly available resource for researchers to design sequencing panels to efficiently target the most mutated genomic regions in their areas of research. Moreover, we demonstrated the ability of a hotSPOT designed sequencing panel to distinguish between normal tissues of varying carcinogen exposure, highlighting the importance of these identified genomic regions for assessing carcinogen exposure and uncovering novel aspects of early carcinogenesis. The hotSPOT tool will lead to more robust experimentation and help us better understand the relationship between mutation burden in healthy and different diseased tissues. 

## Figures and Tables

**Figure 1 cancers-15-01612-f001:**
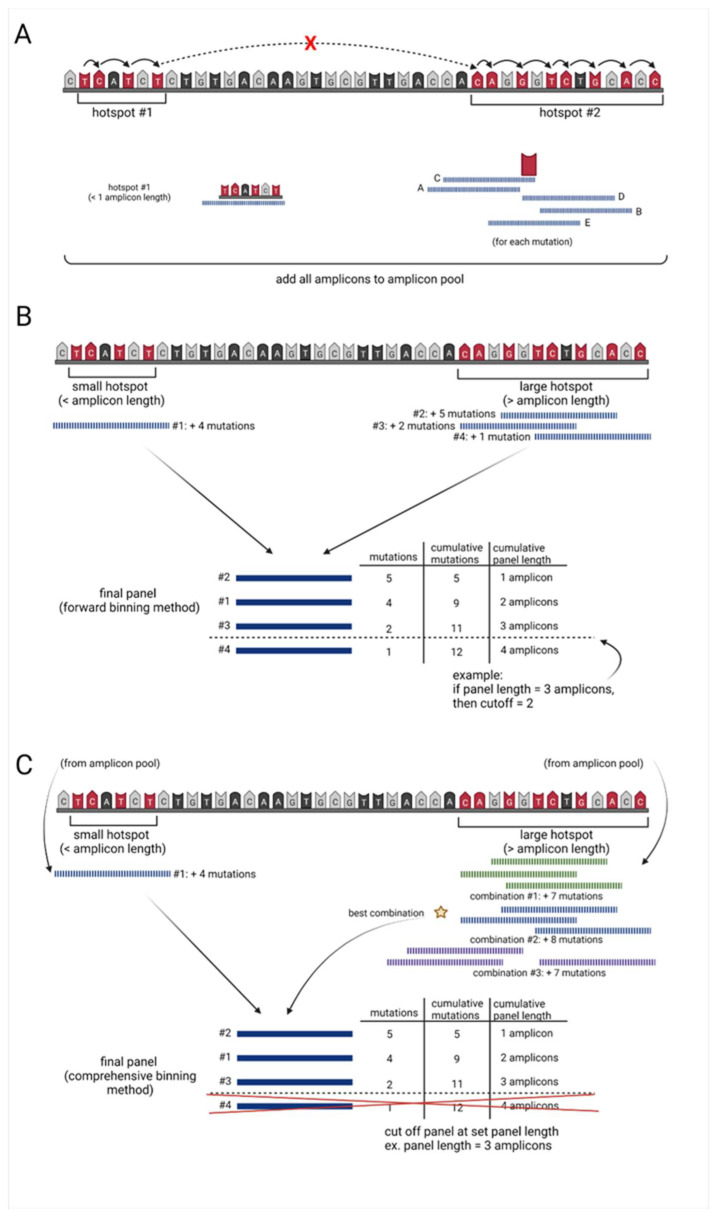
A simplified mechanistic overview of hotspot. (**A**) Amplicon generation, (**B**) forward binning algorithm and (**C**) comprehensive binning algorithm.

**Figure 2 cancers-15-01612-f002:**
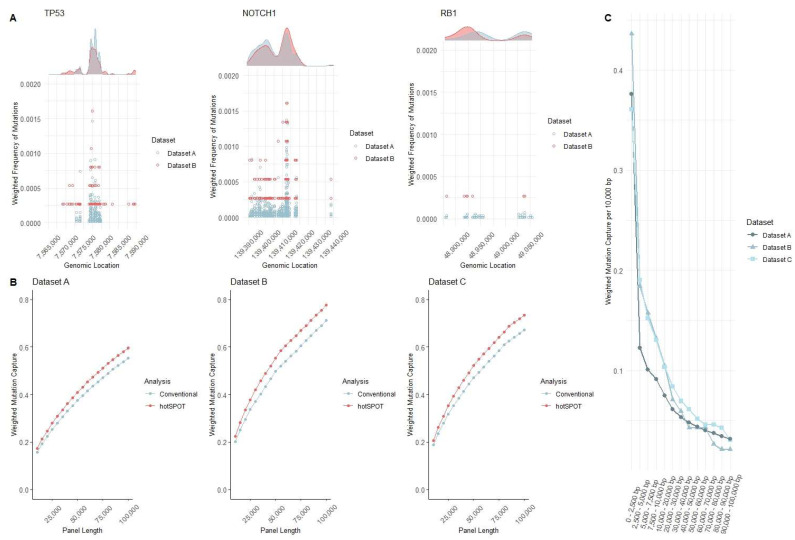
Conceptualization and development of hotSPOT mutation panels. (**A**) Graphical representation of hotspots in frequently mutated (TP53, NOTCH1) and less frequently mutated (RB1) genes in normal skin in datasets A & B. Dots represent the frequency of mutations at each base pair. The plots above represent the density of mutations for each gene. Densities are normalized for each gene based on the average mutation frequency of the datasets. (**B**) Comparison of mutation capture efficacy between conventional and forward hotspot binning methods for Datasets A–C. Mutation capture for each dataset is normalized by the number of mutations present in original analysis. (**C**) Weighted capture efficacy of Datasets A–C ranging from the highest and lowest mutated regions generated by hotSPOT forward binning algorithm.

**Figure 3 cancers-15-01612-f003:**
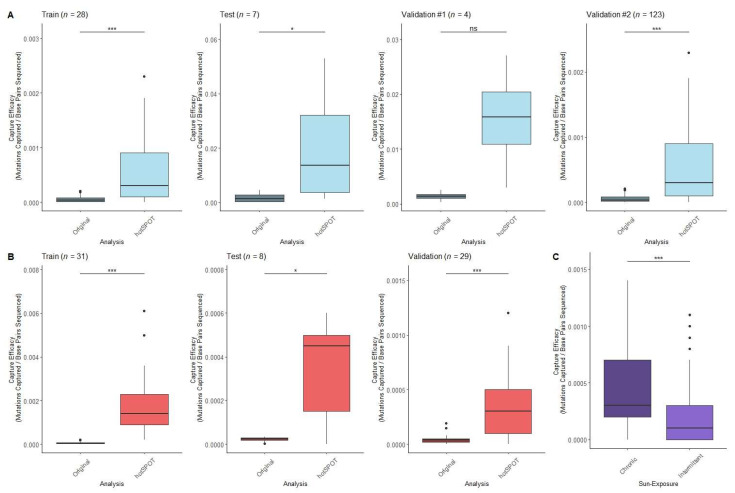
The mutation capture efficiency of hotSPOT for targeted sequencing panel design. (**A**) Difference in capture efficacy from original sequencing analysis compared to 10,000 bp sequencing panel based on training normal epidermis dataset. (**B**) Difference in capture efficacy from original sequencing analysis compared to 10,000 bp sequencing panel based on training cSCC dataset. (**C**) Difference in capture efficacy from chronically sun-exposed and intermittently sun-exposed normal epidermal samples based on 10,000 bp cSCC hotSPOT panel. Statistical significance was calculated using paired Wilcox signed-rank test. Sequencing panels based on forward hotSPOT binning algorithm. (*p* ≥ 0.05 ns, *p* < 0.05 *, *p* < 0.001 ***).

**Figure 4 cancers-15-01612-f004:**
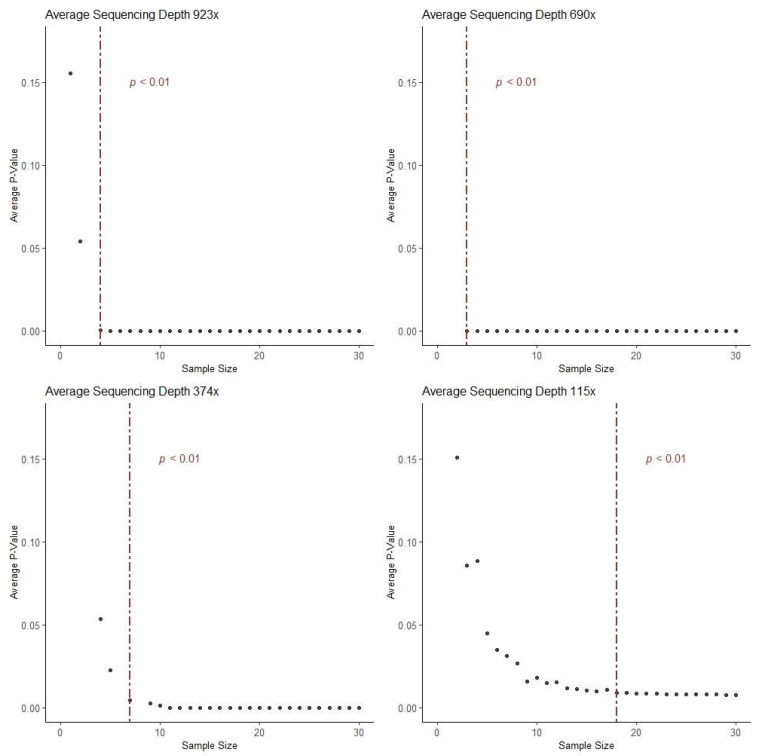
The optimal sample size for effective panel design based on varying sequencing depths 115 – 923×. Statistical significance was calculated using paired Wilcox signed-rank test.

**Table 1 cancers-15-01612-t001:** Summary of human clinically-normal epidermis and cSCC mutation datasets.

Name	Dataset	Source	Samples Sequenced	Average Sequencing Depth	Original Sequencing Panel Size	Sample Type	Panel Design
“Dataset A”	Test/Training Datasets	Fowler et al. [3]	1261	690x	0.39 Mb	Normal Epidermis (Head, forearm, leg, trunk, abdomen)	74 Cancer- related genes
“Dataset B”	Validation Dataset #1	Martincorena et al. [4]	234	374x	0.67 Mb	Normal Epidermis (Eyelid)	74 Cancer- related genes
“Dataset C”	Validation Dataset #2	Hernando et al. [2]	123	923x	0.32 Mb	Normal Epidermis (Back, chest, leg, upper arm, neck, face, hands)	46 genes frequently mutated in skin cancer
	Test/Training Datasets	Pickering et al. [6]	39	115x	-	Tumor	WES
	Validation Dataset	Inman et al. [9]	36	54x	-	Tumor	WES

**Table 2 cancers-15-01612-t002:** The comparison of computation time (seconds) and mutation count for four genes in dataset A, based on hotSPOT algorithms and comprehensive algorithm.

ALGORITHM	FGFR3		FLG2		MLL2		MUC17	
	Time	Count	Time	Count	Time	Count	Time	Count
hotSPOT “Forward”	0.696	45	0.864	14	0.888	27	1.238	45
hotSPOT “Comprehensive”	3.594	45	6.232	14	3.344	27	8.642	45
Exhaustive	34.455	45	298.415	14	10.668	27	569.354	45

## Data Availability

Access to hotSPOT web application is available at https://rpccc-paraghlab-sgrant.shinyapps.io/hotspot/. Access to raw code is available at: https://github.com/sydney-grant/hotSPOT.

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
