# Peer review of "HotSPOT: A Computational Tool to Design Targeted Sequencing Panels to Assess Early Photocarcinogenesis"

_cancers, 2023, doi:10.3390/cancers15051612_

Round 1

Reviewer 1 Report

Remarks to the Author: Sydney R Grant and colleagues desigend a computational algorithm which improved mutation capture efficacy. It is an interesting work research, but there are still some things that need to be revised. Overall, I would support the publication of this study after minor revision. My comments: 1. The introduction need to be rewritten. The point of this article is not what will cause mutation 2. The logic of results and discussions needs to be reorganized. Either the results and discussions are written together, or the results and discussions are written separately.

Author Response

Reviewer #1

Thank you for reviewing our manuscript and for your helpful critiques. We made the recommended modifications.

Comments and Suggestions for Authors

  1. The introduction need to be rewritten. The point of this article is not what will cause mutation.

The introduction has been thoroughly revised to highlight the paper's focus on identifying which genomic areas most sensitive to mutations rather than what may cause these mutations. 

  1. The logic of results and discussions needs to be reorganized. Either the results and discussions are written together, or the results and discussions are written separately.

The results and discussion sections of this paper have been combined.

Reviewer 2 Report

The authors present a very comprehensive review of the most recent literature on photocarcinogenesis and skin cancer.

I feel that the paper can be accepted in the current form. 

Author Response

Reviewer #2: 

Thank you very much for reviewing our manuscript.

Comments and Suggestions for Authors

The authors present a very comprehensive review of the most recent literature on photocarcinogenesis and skin cancer. I feel that the paper can be accepted in the current form.

Reviewer 3 Report

The authors of HotSPOT: a computational tool to design targeted sequencing panels to assess early photocarcinogenesis developed a very interesting computational tool for designing efficient sequencing target panels.

If you allow me, I suggest extending introduction and discussion sections with more data about skin cancer, etiologic factors (e.g., oncogenic viruses), possible targeted therapies.

I would suggest, if allow me, the cite and discuss the next connected papers:

      ·       DOI: 10.1186/s13027-022-00472-w

·       DOI: 10.1016/j.jid.2020.04.028

·       DOI: 10.1016/j.pvr.2019.04.003

·       DOI: 10.3390/pathogens11040479

In my view, in this way, it could be increased the addressability of this study to more readers, including clinicians.

Thank you.

Author Response

Reviewer #3: 

Thank you for reviewing our manuscript and for your thoughtful suggestion.

Comments and Suggestions for Authors

If you allow me, I suggest extending introduction and discussion sections with more data about skin cancer, etiologic factors (e.g., oncogenic viruses), possible targeted therapies. I would suggest, if allow me, the cite and discuss the next connected papers.

We restructured the introduction and highlighted the skin cancer relevance of the work. However, adding more information on skin cancer etiologic factors and targeted therapies in skin cancer we feared would have diverted attention from the main subject of the manuscript. Nevertheless, based on your comment in our follow-up paper addressing specific hotspots and their significance will better connect the dots between etiologic factors and therapies.

Reviewer 4 Report

In their manuscript the authors present a computational tool (hotSPOT) that should help to simplify designing efficient sequencing panels fitted to genomic hotspot areas. The hotSPOT algorithm is generic and may be used to increase mutation capture efficacy in applications. The manuscript describes an application of hotSPOT in a dermatological setting to identify mutation profiles in epidermal tissues from “normal” skin. Data sets used in this hotSPOT application have already been published by other groups.

Comments:

-       After reading the Introduction, it has not become clear what the primary goal of the manuscript is. Is the focus on delineating the clustering of mutations in genomic hotspots in sun-exposed normal skin? Or is the presentation of the hotSPOT software for identifying targeted sequencing panels the primary intention? When progressing to the Methods section it is evident that the computational aspects are the focus of this manuscript. I doubt that Cancers is the right journal for such a computational topic.

-       Little detail about the data sets used in the application are given by the authors (readers have to go to the references cited).

-       What was the rationale to use the two-sample Kolmogorov-Smirnov test for comparing two mutation distributions? The KS test has a very limited statistical power, other nonparametric statistical tests have been shown to be superior. In addition, the authors claim to have used the “paired Mann-Whitney U test”. But for what? In addition, there is no such statistical test, the Mann-Whitney U test is (only) suited to fit to an unpaired two-sample situation. It is nice to get the complete information which R packages have been used in the computation, but this degree of detail is definitely lacking in section 2.6 describing the statistical analyses. Details of the design and specific analysis of the evaluation study to show an increase in mutation capture efficacy when applying hotSPOT are not described in the Methods.

-       Many important results of the evaluation can be found only in the Online Supplement Figures. Shouldn’t this be part of the main test?

Overall, the manuscript is a nice read, but is ill-placed in the journal Cancers due to its purely computational focus. The description of the hotSPOT application in the dermatological setting to demonstrate the efficacy gain due to hotSPOT is not well-structured and lacks necessary details. This part (which would justify Cancers as the publication venue if it were a strong part) has no implications for understanding the transition from normal epidermal tissue to skin cancer (it is a mere illustration of applying hotSPOT)

Author Response

Reviewer #4: 

Thank you for reviewing our manuscript and for all your recommendations aimed at improving our manuscript. We addressed all your comments. Please see the details below.

Comments and Suggestions for Authors

  1. After reading the Introduction, it has not become clear what the primary goal of the manuscript is. Is the focus on delineating the clustering of mutations in genomic hotspots in sun-exposed normal skin? Or is the presentation of the hotSPOT software for identifying targeted sequencing panels the primary intention? When progressing to the Methods section it is evident that the computational aspects are the focus of this manuscript. I doubt that Cancers is the right journal for such a computational topic.

The introduction has been revised to highlight the paper’s focus on presenting hotSPOT as a tool for identifying targeted sequencing panels and clarified that we needed hotSPOT for addressing our specific questions about normal skin and skin cancer mutation patterns To highlight the latter  we now include an additional analysis showing the use of a hotSPOT targeted sequencing panel to measure photocarcinogenic progression in the normal-appearing epidermis.

  1. Little detail about the data sets used in the application are given by the authors (readers have to go to the references cited).

The methods section describing the publicly available datasets used has been completely revised to include further detail on the quantity, patient number, sampling method, and size of epidermis used for sequencing experiments in each dataset.

  1. What was the rationale to use the two-sample Kolmogorov-Smirnov test for comparing two mutation distributions? The KS test has a very limited statistical power, other nonparametric statistical tests have been shown to be superior. In addition, the authors claim to have used the “paired Mann-Whitney U test”. But for what? In addition, there is no such statistical test, the Mann-Whitney U test is (only) suited to fit to an unpaired two-sample situation. It is nice to get the complete information which R packages have been used in the computation, but this degree of detail is definitely lacking in section 2.6 describing the statistical analyses. Details of the design and specific analysis of the evaluation study to show an increase in mutation capture efficacy when applying hotSPOT are not described in the Methods.

Thank you for pointing out the inaccuracies and insufficient details in the statistics session. The methods section describing the statistical analyses has been extended to include more detail. Kolmogorov-Smirnov test was used to demonstrate the lack of statistical difference between mutation distribution of different epidermal samples based on bioinformatics and biostatistics advice as it is sensitive to differences in both the location and the shape of the empirical cumulative distribution functions. The methods behind this analysis are now included in greater detail. Paired Wilcox signed-rank test has been utilized for the determination of the statistical significance of the mutation capture increase using hotSPOT.

  1. Many important results of the evaluation can be found only in the Online Supplement Figures. Shouldn’t this be part of the main test?

Thank you for pointing this out. Based on your recommendations several supplemental figures and tables have been moved to be a part of the main text.

Reviewer 5 Report

The manuscript describes a computational tool termed hotSPOT aimed at simplifying the design of efficient sequencing panels fitted to genomic hotspot areas. The algorithm is generic and could be applied in different settings, but the manuscript shows an application of the tool to identify epidermal mutation profiles in clinically normal-appearing skin based on an already published data set (split into learning and validation subsamples) and two other independent validation data sets that also have already been used in other publications. If I understand the authors correctly, their primary aim was to demonstrate an efficiency gain when using hotSPOT compared to other approaches to design sequencing panels.

Specific remarks:

-       The Methods section is used (only) to describe the hotspot algorithm step-by-step. Details of the design of the evaluation study to demonstrate superiority of hotSPOT are not described here. Quite unusual for a scientific paper, the Results section contains a description of what has been done in the evaluation, while the quantitative results of the evaluation are mostly presented in supplemental figures that are not part of the main manuscript. Probably I am little bit old-fashioned, but I do not feel much sympathy for such a structure.

-       The data sets used are not well described in the manuscript. Without inspecting cited references the reader gets no information how the cases entering the data sets have been chosen (eligibility criteria, recruitment procedures, definition of “clinically normal-appearing skin” etc.). A brief summary extending the technical facts in Supplementary Table 1 would be helpful.

-       I did not get the rationale why the evaluation focused only on TP53, NOTCH1, and RB1 as there several other genes like e.g. CDKN2A (and others) that have been established as “skin cancer genes” for years and would be natural targets. Did the authors intend to provide a proof-of-concept study? The gain in efficiency will not be independent from the gene studied, thus an extension of the number of genes will lead to more information on the properties of the algorithm when applied in this setting of skin cancer genes.

Overall, I join the authors in their claim that more efficient sequencing panels capturing high mutation genomic regions are urgently needed and that the hotspot algorithm may provide a reasonable step into that direction. The manuscript is, however, more a generic description of the computational aspects of the algorithm and has only limited implications for studying the transition from normal epidermal tissue to skin cancer as it uses skin cancer as a mere illustration and not as the focus of research presented. Therefore, a journal with a more computational focus would be a better venue for this manuscript.

Author Response

Review #5: 

Thank you for reviewing our manuscript and for all your recommendations aimed at improving our manuscript. We addressed all your comments. Please see the details below.

Comments and Suggestions for Authors

  1. The Methods section is used (only) to describe the hotspot algorithm step-by-step. Details of the design of the evaluation study to demonstrate superiority of hotSPOT are not described here. Quite unusual for a scientific paper, the Results section contains a description of what has been done in the evaluation, while the quantitative results of the evaluation are mostly presented in supplemental figures that are not part of the main manuscript. Probably I am little bit old-fashioned, but I do not feel much sympathy for such a structure.

The methods and results sections have been revised to include more detail in the methods section and limit the discussion of methods within the results.

  1. The data sets used are not well described in the manuscript. Without inspecting cited references the reader gets no information how the cases entering the data sets have been chosen (eligibility criteria, recruitment procedures, definition of “clinically normal-appearing skin” etc.). A brief summary extending the technical facts in Supplementary Table 1 would be helpful.

The methods section describing the publicly available datasets used has been revised to include further detail on the quantity, patient number, sampling method, and size of epidermis used for sequencing experiments in each dataset.

  1. I did not get the rationale why the evaluation focused only on TP53, NOTCH1, and RB1 as there several other genes like e.g. CDKN2A (and others) that have been established as “skin cancer genes” for years and would be natural targets. Did the authors intend to provide a proof-of-concept study? The gain in efficiency will not be independent from the gene studied, thus an extension of the number of genes will lead to more information on the properties of the algorithm when applied in this setting of skin cancer genes.

The focus on TP53, NOTCH1, and RB1 was intended to provide visual examples of the similarities in mutation distribution between different datasets in both highly mutated genes in skin cancer, and even in less mutated genes in skin cancer. This section of the text has been modified to include more detail on this reasoning. CDKN2A and many other well know skin cancer genes are more relevant for melanoma than keratinocyte carcinoma. Given the paucity of melanocytes in normal skin. Melanocyte mutational signatures cannot be studied on normal epidermal samples without melanocyte enrichment. We have proposals aimed at studying mutations in melanocytes from normal skin, and in those projects, CDKN2A will be one of the genes of focus.

Round 2

Reviewer 4 Report

The revised manuscript was expanded and partially corrected. The subsection describing the statistical analysis particularly benefited from revision. Restructuring of the text in the other sections has improved clarity and readability. I remain skeptical that the computational focus of the manuscript is appropriate for the journal Cancers, but this aspect must be decided by the editor.

Reviewer 5 Report

The authors constructively addressed my comments and submitted a revised manuscript that incorporated the requested information and reorganized the presentation. I am still not convinced that the journal Cancers is the right journal for a manuscript with such a strong computational focus but this is an editorial decision and not mine to make.